# Bacillus simplex treatment promotes soybean defence against soybean cyst nematodes: A metabolomics study using GC-MS

Wen-shu Kang[1], Li-jie Chen[2], Yuan-yuan Wang[3], Xiao-feng Zhu[2], Xiao-yu Liu[4], Hai-yan Fan[2], Yu-xi Duan[2]*

1 College of Environment, Shenyang University, Shenyang, China, 2 College of Plant Protection, Shenyang Agricultural University, Shenyang, China, 3 College of Biotechnology, Shenyang Agricultural University, Shenyang, China, 4 College of Sciences, Shenyang Agricultural University, Shenyang, China

* duanyx6407@163.com

**Data Availability Statement:** All relevant data are within the manuscript and its Supporting Information files.

## Abstract

We aimed to profile the metabolism of soybean roots that were infected with soybean cyst nematodes and treated with *Bacillus simplex* to identify metabolic differences that may explain nematode resistance. Compared with control soybean roots, *B. simplex*-treated soybean roots contained lower levels of glucose, fructose, sucrose, and trehalose, which reduced the nematodes' food source. Furthermore, treatment with *B. simplex* led to higher levels of melibiose, gluconic acid, lactic acid, phytosphingosine, and noradrenaline in soybean roots, which promoted nematocidal activity. The levels of oxoproline, maltose, and galactose were lowered after *B. simplex* treatment, which improved disease resistance. Collectively, this study provides insight into the metabolic alterations induced by *B. simplex* treatment, which affects the interactions with soybean cyst nematodes.

## Introduction

Soybean seeds contain high protein and oil content, making soybean a vital crop for the industry and agriculture. This places a high importance on efficient production of soybean with a high yield. *Heterodera glycines*, also known as soybean cyst nematode (SCN), is a major soybean pathogen that can hinder the growth and production of soybean. Environmentally friendly biological control methods are becoming increasingly popular to prevent nematode diseases to improve the sustainability of agriculture. Plant growth-promoting rhizobacteria (PGPR) colonise the rhizosphere of various species of plants and are now used as a biological control tool to increase growth and enhance disease resistance to fungi, bacteria, viruses, and nematodes [1]. Our past studies have discovered that the Sneb545 strain of *Bacillus simplex* enhances soybean resistance to SCN [2]. Coating seeds with these bacteria is simple and produces a cost-effective strategy to control SCN. This priming method leads to quicker and more effective development of active defence responses against phytopathogens without negatively affecting soybean plant growth [3].

**Funding:** This work was supported by the National Natural Science Foundation of China (grant number: 313300630) and China Agriculture Research System (No. CARS-04-PS13).

**Competing interests:** The authors have declared that no competing interests exist.

An increasing number of reports have shown that metabolomics research using gas chromatography-mass spectrometry (GC-MS) is advantageous for studying plant-nematode interactions [4–6]. GC-MS robustly identifies and quantifies the metabolites from plant extracts [7, 8] and elucidates the primary pathways used in metabolism while offering good sensitivity and reliability. It is significantly more sensitive than nuclear magnetic resonance and is more reliable compared with liquid chromatography–linked mass spectrometry [9].

In this study, we used GC-MS to profile the metabolites of soybean roots after treatment with Sneb545 and investigated the metabolic changes during SCN infection. To the best of our knowledge, we are the first to systematically analyse the metabolic changes after Sneb545-treatment in SCN-infected soybean roots. We hypothesized that bacteria could improve soybean resistance to nematode infection by inducing the soybean plant roots to produce substances with nematode killing activity or inhibiting the nematode's growth and development.

## Methods

### Bacterial strains

We used the Sneb545 strain of *B. simplex*, which was previously selected in a two-year field experiment by the Northern Nematode Institute of Sheng yang Agriculture University in China [2].

### Plant growth, treatment, and harvest

Seeds of a susceptible soybean cultivar, Liao15, were sterilised in 1% sodium hypochlorite and moderately shaken for 3 min as described previously [10]. Subsequently, the seeds were washed with 70% ethanol and 1% of the seeds were coated with Sneb545. Control seeds were treated with sterile distilled water. We potted the sterilized seeds in a 1:1 mixture of sterilised soil and sand and grew the plants in a 16:8 light-dark cycle at $26 \pm 3°C$.

The race 3 strain of SCN was collected from soil samples by washing with a 1.9 M sucrose solution followed by centrifugation at 2,000 rpm. SCN cysts were incubated in a chamber dip at 26°C in 3 mM ZnSO4 until second-stage juveniles (J2) hatched from the eggs. The J2 nematodes were pooled in fresh distilled water and their concentration was determined. The roots of untreated and Sneb545-treated seedlings were inoculated with SCN by exposure to 1 ml J2 solution at a concentration of 1,000 J2/ml mixed with 0.2% agar when two leaves had emerged from the seedlings. The respective untreated and Sneb545-treated controls were not inoculated with SCN. When two true leaves were grown from soybean, the root of soybean was inoculated with soybean cyst nematode. Root samples of two biological replicates per treatment were harvested in triplicate at 5 and 10 days post-inoculation (dpi) and were snap-frozen in liquid nitrogen and stored at –80°C for further analyses.

### Nematode infection assay

The juvenile SCN in Sneb545-treated and control Liao15 roots were examined over time and we detected the development of SCN from J2 to J3 stage according to Bird's protocol by staining the SCN with 0.01% acid fuchsin as described previously [11].

### Extraction and derivation of metabolites

We added 0.4 ml of methanol and water mixture in a 3:1 volume ratio to 50 mg of soybean root tissue. Then, 20 μl 2 mg/ml Adonitol in distilled water was added as an internal standard. A ball mill was used to homogenise the mixture for 4 min at 40 Hz followed by ultrasound treatment for 5 min in ice water. The samples were then centrifuged for 15 min at 12000 rpm

and 4˚C. We transferred 0.35ml of the supernatant to 2 ml GC-MS glass vials and pooled 9 μl from each sample for quality control checks. The liquid was evaporated in a vacuum concentrator at room temperature. The samples were dissolved in 20 μl 20 mg/ml methoxyamine hydrochloride in pyridine and incubated for 30 min at 80˚C. Subsequently, 30 μl 1% TMCS in trimethylchlorosilane (BSTFA) was added to the samples and further incubated for 2 h at 70 ˚C. Finally, 7 μl of fatty acid methyl ester mixture in chloroform was added to the quality control sample. The samples were mixed before GC-MS analysis.

## GC-MS analysis

An Agilent 7890 gas chromatograph (Agilent, USA) was used in conjunction with a Pegasus® HT time-of-flight mass spectrometer (LEGO, USA) for gas chromatography time-of-flight mass spectrometry (GC/TOFMS) analyses. Our setup used a 30 m × 250 μm inner diameter DB-5MS column with a film thickness of 0.25 μm (J&W Scientific, USA). We injected 1 μl of the sample for analysis. The carrier gas was helium with a purge flow of 3 ml/min and a gas flow of 1 ml/min. The temperature cycle was as follows: 50 ˚C for 1 min, which was then increased to 300 ˚C at a rate of 10˚C/min, and finally 300˚C for 8 min. The temperatures of the injection, transfer line, and ion source were maintained at 280, 270, and 220˚C, respectively and the energy was set at -70 eV. After a solvent delay of 460 s, we acquired the data with an m/z range of 50–500 at a rate of 20 spectra/s.

## Nematode mortality assessment

We assessed the mortality rate of the SCNs when exposed to 500 μg/ml phytosphingosine and noradrenaline dissolved in 2% Tween 80 (Sigma-Aldrich, USA) and 4% methanol (Sigma-Aldrich, USA) as described previously [12]

## Statistical analysis

The nematode infection and mortality data were examined using Student's t-test in SPSS 17.0 (IBM, USA) and a P-value of $< 0.05$ was considered statistically significant. Chroma TOF4.3X (LECO Corporation, USA) and the LECO-Fiehn Rtx5 database were used to exact, align, and identify the peaks and filter and calibrate the baseline data. Furthermore, we also performed deconvolution analysis and determined the peak area as described previously [12]. The peak data were analysed by SIMCA14.1 (Umetrics, Sweden) for orthogonal projections to latent structures-discriminate analysis (OPLS-DA).

# Results

## Developmental differences of juveniles in differentially treated soybean roots

Sneb545-treated and control soybean roots were inoculated with SCN and we examined the SCN development. The number of J2 SCN at all time points were significantly lower with Sneb545 treatment than in the controls (Fig 1).

## Metabolic profile

Total ion chromatograms of soybean roots samples revealed that 15 metabolites were confirmed from 411 peaks and included sugars, alcohols, and organic acids (Fig 2).

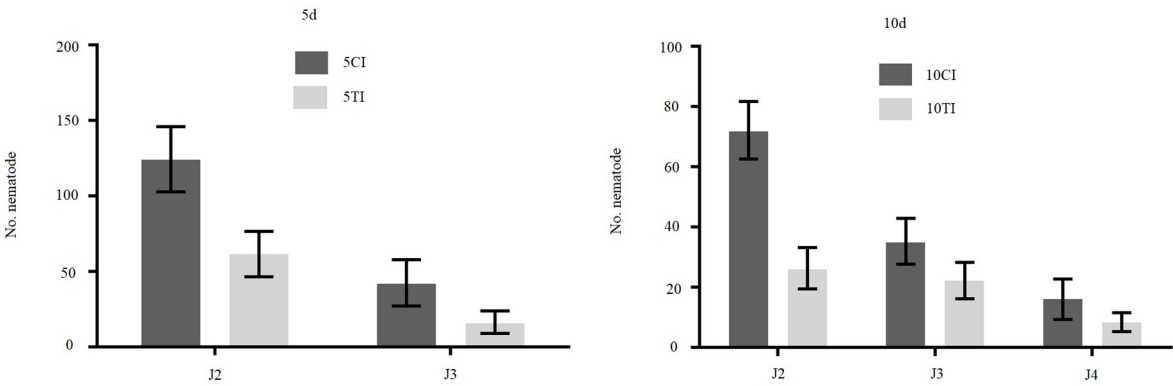

**Fig 1. The different stages of SCN in mock and Sneb545- treated soybeans roots at days 5 and 10.** 5CI/10CI indicate mock-treated soybean roots and 5CI/10CI indicate Sneb545-treated soybean roots at days 5 and 10 after SCN inoculation respectively.

## Metabolic profiling of SCN-infected soybean roots treated with Sneb545

OPLS-DA was used to examine the diverse metabolic patterns from the chromatograms. According to the R2 and Q2 values, the models were of good quality and representative of the data (Table 1). Soybeans treated with Sneb545 and control soybeans were distinguishable in the OPLS-DA plot (Fig 3).

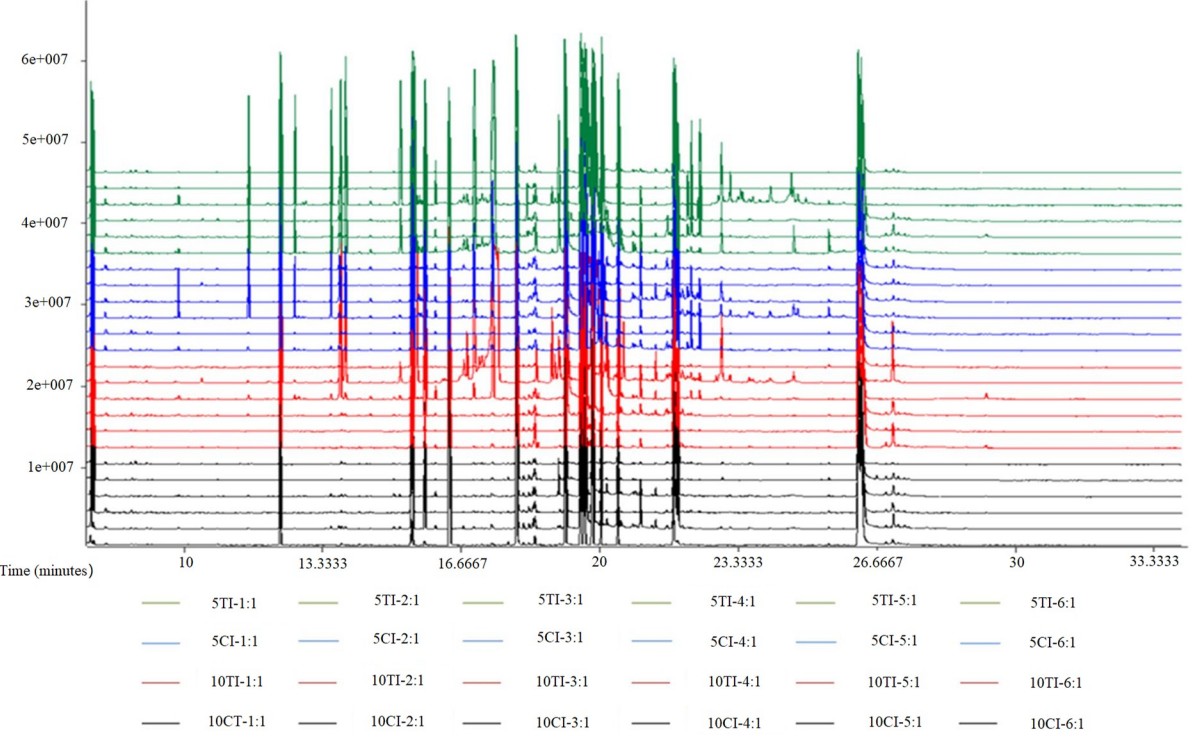

**Fig 2. GC/MS TICs of soybean roots samples.** The y-axis indicates the relative mass abundance and the x-axis represents the retention time.

**Table 1. Assessment of modelling quality.**

| Group | R2X | R2Y | Q2 |
|---|---|---|---|
| 5TI VS 5CI | 0.222 | 0.984 | 0.262 |
| 10TI VS 10CI | 0.328 | 0.943 | 0.18 |

### Potential sugar metabolic pathway markers for Sneb545-induced soybean resistance to SCN

We used the variable importance in the projection (VIP) of the first principal component of the OPLS-DA model described above (threshold > 1) as well as the P-value of the Student's t-test (threshold < 0.1) to select the variables that contributed to group separation (Table 2). We found that treating soybeans with Sneb545 resulted in downregulation of L-threose, allose, xylose, fructose, D-talose, galactose, phytosphingosine, sucrose, trehalose, and gentio-biose while lactic acid, gluconic acid, noradrenaline, phytosphingosine, and melibiose were upregulated.

### SCN mortality assessment

Assessment of SCN mortality indicated that phytosphingosine and noradrenaline effectively induced SCN death (Table 3), showing mortality rates of 79.5% vs 84.0% and 70% vs 79% at 24 h and 48 h, respectively.

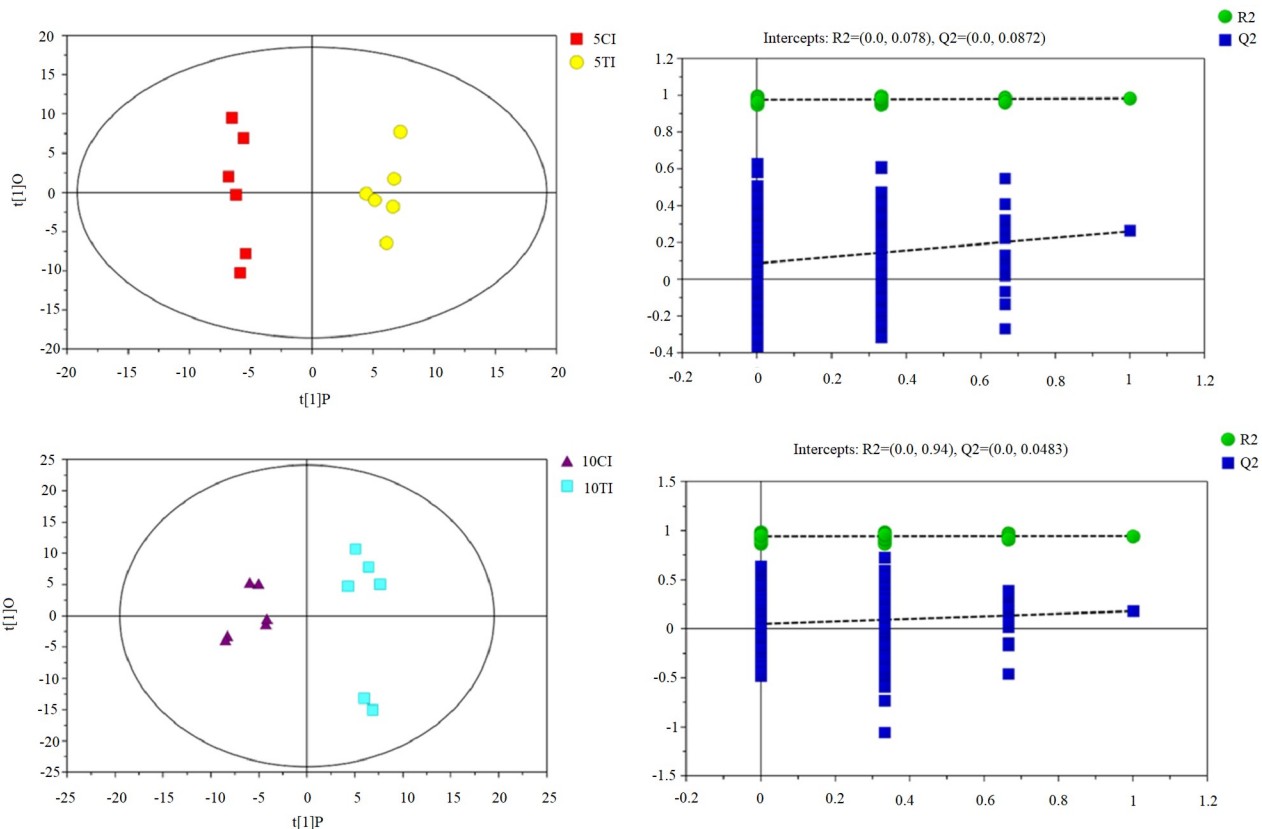

**Fig 3. OPLS-DA model analysis performed on the most diverse sample at (A) day 5 and (B) day 10 post-inoculation.** CI indicates SCN-inoculated mock-treated soybean roots and TI indicates SCN-inoculated Sneb545-treated soybean roots.

**Table 2. List of different metabolites found in the Sneb545-treated and control soybean roots.**

| ID | Peak | RT | Count | Mass | VIP | p-Value | Fold change |
|---|---|---|---|---|---|---|---|
| 54 | Lactic acid[a] | 9.21031,0 | 56 | 117 | 1.58633 | 0.094723 | 1.438946963 |
| 279 | Oxoproline[a] | 15.7083,0 | 48 | 156 | 1.59914 | 0.094283 | 0.258598133 |
| 444 | Fructose [a] | 19.667,0 | 14 | 465 | 1.77095 | 0.076189 | 4.41631E-05 |
| 464 | Galactose [a] | 20.0351,0 | 44 | 160 | 2.5944 | 0.00028 | 0.132678593 |
| 497 | Gluconic acid[b] | 20.8502,0 | 11 | 333 | 1.38921 | 0.094828 | 2.708045847 |
| 548 | Noradrenaline[b] | 22.2533,0 | 55 | 84 | 1.69914 | 0.097282 | 2.081636721 |
| 661 | Phytosphingosine[b] | 25.7827,0 | 29 | 84 | 1.81026 | 0.048436 | 6584.635105 |
| 675 | Sucrose[a] | 26.2919,0 | 12 | 66 | 1.77 | 0.076825 | 3.44555E-06 |
| 699 | Trehalose[b] | 27.0453,0 | 30 | 191 | 1.7515 | 0.025149 | 0.172619279 |
| 703 | Maltose[b] | 27.1762,0 | 42 | 160 | 2.25554 | 0.049032 | 3.39185E-05 |
| 725 | Melibiose [b] | 28.0361,0 | 45 | 73 | 1.83427 | 0.077231 | 7.49023791 |

[a]Significant difference at 5 dpi.

[b]Significant difference at 10 dpi.

## Discussion

In this study, the number of SCN in different stages at 5 dpi was comparable to our previous research [13]. Based on our analyses using GC-MS metabolic profiling, we found that Sneb545 treatment of soybean promoted metabolic resistance and identified 15 metabolites that contributed to SCN resistance.

The sedentary endoparasite cyst nematode, *Heterodera schachtii*, develops a highly specific and intricate host-pathogen interaction, which results in the formation of a syncytial feeding site in the root vascular tissue [14]. The cell wall between the initial syncytial cell and neighbouring cells gradually disappears to form a syncytial cell complex [13, 14]. Studies have discovered that sedentary cyst-forming nematodes directly access the phloem and feed on the sucrose that is present [15], which has shown to be the major source of carbohydrates in *Arabidopsis thaliana* roots [16]. *H. schachtii* juveniles are also dependent on sucrose supplied by transporters during feeding-site induction and establishment [17]. Invertases catalyse the hydrolysis of sucrose into fructose and glucose, which can be used by nematodes as a food source [18]. Our results indicated that sucrose, fructose, and trehalose were dramatically reduced in the Sneb545-treated soybean roots, suggesting that Sneb545 treatment reduces the nutrient source for nematodes and inhibits their development.

In addition to reducing the nutritional source of the nematodes, Sneb545 induced soybean roots to produce substances with nematocidal activity. There has been no direct evidence of nematocidal activity by melibiose and gluconic acid. However, these two substances are known to be present in different extracts that possess nematocidal activities [19–21]. We demonstrated that melibiose and gluconic acid content were elevated in the Sneb545-treated group at 10 dpi, which may have enhanced nematocidal activity. Lactic acid has been shown to harbour nematocidal activity against plant-parasitic, free-living, and predacious nematodes [22]

**Table 3. Mortality of J2 SCN at 24 and 48 h after exposure to 500 μg/ml phytosphingosine and noradrenaline.**

| Treatment | 24 h corrected J2 mortality % (mean ± SE) | 48 h corrected J2 mortality % (mean ± SE) |
|---|---|---|
| Phytosphingosine | 79.50 ± 0.18 | 84 ± 0.29 |
| Noradrenaline | 70 ± 0.19 | 79 ± 0.27 |

and has also been isolated from the culture filtrate of the YS1215 strain of *Lysobacter capsica*, showing nematocidal potential to control root-knot nematode [23]. Furthermore, we observed that the Sneb545 treatment group showed increased lactic acid content, thus possibly impeding nematode development. Phytosphingosine is a major sphingoid base from fungi and induces apoptosis in *Aspergillus nidulans* that is similar to the caspase-independent apoptosis observed in mammalian systems [24]. Phytosphingosine isolated from *Bacillus cereus* strain S2 shows nematocidal activity against *Meloidogyne incognita* and *Caenorhabditis elegans* [25]. We found 6,584-fold higher phytosphingosine content in the Sneb545-treated group compared with the control group, suggesting greatly enhanced resistance to SCN. Noradrenaline mediates responses to acute stress that result in decreased immune responses in mammals [26]. These chemical cues in the nervous system of *C. elegans* show comparable consequences for the immune system [27]. We demonstrated in our study that phytosphingosine and noradrenaline had nematocidal activity at a concentration of 500 μg/ml. Therefore, Sneb545-induced noradrenaline may disrupt the nervous system and immune system balance to promote an abnormal immune function in SCN and may disturb normal growth and development.

In addition to the effects outlined above, Sneb545 also improved the disease resistance of the plant to SCN. Studies have reported alternations in the metabolites of NIL 34–23 (resistant haplotype) and NIL 34–3 (susceptible haplotype) seeds due to SCN infection. Similarly, oxoproline, maltose, and galactose are lower in resistant soybean than in susceptible soybean [28] and agrees with our findings. Therefore, we speculate that Sneb545 treatment of soybean produces some qualities of SCN disease-resistant soybean variants.

In summary, differentially expressed metabolites were found between Sneb545-treated soybean roots and mock-treated soybean roots that had been inoculated with SCN. These metabolites improved the resistance of soybean to SCN in two ways by reducing nematode food sources and producing substances with nematocidal activity.

## Supporting information

**S1 Table. List of different metabolites in CI group at 5dpi.**
(XLSX)

**S2 Table. List of different metabolites in TI group at 5dpi.**
(XLSX)

**S3 Table. List of different metabolites in CI group at 10dpi.**
(XLSX)

**S4 Table. List of different metabolites in TI group at 10dpi.**
(XLSX)

## Acknowledgments

We would like to thank the teachers and students at Shenyang Agricultural University for their guidance and help. We are particularly grateful to Prof. Yuxi Duan for his assistance.

## Author Contributions

**Funding acquisition:** Yu-xi Duan.

**Supervision:** Li-jie Chen, Yuan-yuan Wang, Xiao-feng Zhu, Xiao-yu Liu, Hai-yan Fan.

**Writing – review & editing:** Wen-shu Kang.

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
