## [Decision Letter · Decision Letter 0]

19 May 2020

PONE-D-20-07320

Bacillus simplex treatment promotes soybean defense against soybean cyst nematodes: A metabolomics study using GC-MS

PLOS ONE

Dear Dr. Kang,

Thank you for submitting your manuscript to PLOS ONE. After careful consideration, we feel that it has merit but does not fully meet PLOS ONE’s publication criteria as it currently stands. Therefore, we invite you to submit a revised version of the manuscript that addresses the points raised during the review process.

Please also arrange professional English editing for this manuscript to improve its language clarity.

We would appreciate receiving your revised manuscript by Jul 03 2020 11:59PM. To enhance the reproducibility of your results, we recommend that if applicable you deposit your laboratory protocols in protocols.io, where a protocol can be assigned its own identifier (DOI) such that it can be cited independently in the future. For instructions see: http://journals.plos.org/plosone/s/submission-guidelines#loc-laboratory-protocols

We look forward to receiving your revised manuscript.

Kind regards,

Hon-Ming Lam, Ph.D.

Academic Editor

PLOS ONE

Reviewers' comments:

Reviewer's Responses to Questions

**Comments to the Author**

1. Is the manuscript technically sound, and do the data support the conclusions?

Reviewer #1: Yes

Reviewer #2: Yes

2. Has the statistical analysis been performed appropriately and rigorously? 

Reviewer #1: Yes

Reviewer #2: No

3. Have the authors made all data underlying the findings in their manuscript fully available?

Reviewer #1: Yes

Reviewer #2: Yes

4. Is the manuscript presented in an intelligible fashion and written in standard English?

Reviewer #1: Yes

Reviewer #2: No

5. Review Comments to the Author

Reviewer #1: The article : Bacillus simplex treatment promotes soybean defense against soybean cyst

nematodes: A metabolomics study using GC-MS is well written. The content is easy to follow and the methodologies are clearly stated. However, feature extractions of metabolites (relatively low coverage nowadays but acceptable) could be improved.

Reviewer #2: This manuscript reports the results from a pot experiment and revealed the effects of Bacillus inoculation on root metabolic changes in soybean plants infected with cyst nematodes (SCN). The findings showed that the bacterial inoculation functioned in three different aspects: reduced food source, produced nematocidal substances and improved disease-resistance ability. The main concern is that only one soybean cultivar, one bacterial strain, and one isolate of cyst nematode were used in this study. Another concern is the language writing which needs further improvements throughout the manuscript (lots of typos and grammatical errors). Some other specific comments:

L. 47-50: the objective is quite general. It would be good to add hypotheses of this study.

L. 59: not clear of how the bacteria inoculum were coated to the seeds

L. 67: not clear of the seedling age or growth stage at the time of inoculation with SCN

L. 70: plants were assessed at 5 and 10 days after inoculation: not sure if such short period was enough for observing interactions between the bacteria and SCN

L. 81: samples of roots?

L. 74: ‘Liaodou15’ here, but ‘Liao15’ in L. 58. Use one and be consistent throughout the manuscript

L. 77-78: last sentence on data analysis can be removed to the “Data analysis” section

L. 107-114: put this section last, i.e. before the ‘Results’ section to include all acquired data in this study in addition to the spectrometry data, such as ANOVA for nematode numbers etc.

L. 129: significant level? The statistical analysis was not mentioned in the M&M section.

L. 130-131: delete this sentence and move it to Discussion section.

6. PLOS authors have the option to publish the peer review history of their article (what does this mean?). If published, this will include your full peer review and any attached files.

Reviewer #1: No

Reviewer #2: No

---

## [Author Response · Author response to Decision Letter 0]

18 Jun 2020

Response to Reviewers

Dear Editors and Reviewers:

Thank you for your letter and for the reviewers’s comments concerning our manuscript entitled “Bacillus simplex treatment promotes soybean defence against soybean cyst nematodes: A metabolomics study using GC-MS”(PONE-D-20-07320). Those comments are all valuable and very helpful for revising and improving our paper, as well as the important guiding significance to our researches. We have studied comments carefully and have made correction which we hope meet with approval. Revised portion are marked in red in the paper. The main corrections in the paper and the responds to the editors and reviewer’s comments are as flowing:

Responds to the editor’s comments:

1. We have found the professional English editing for this manuscript to improve its language clarity. And we have different treatment manuscript labeled followed the 'Response to Reviewers', 'Revised Manuscript with Track Changes' and 'Manuscript'.

2. We have deposited our laboratory protocols in protocols.io.

DOI: dx.doi.org/10.17504/protocols.io.bhbpj2mn

3. We have added the first author’s ORCID.

Responds to the reviewer’s comments:

Reviewer 2:

1. Response to comment: L. 47-50: the objective is quite general. It would be good to add hypotheses of this study.

Reply: We added the hypotheses of this study that “We hypothesized that bacteria could improve soybean resistance to nematode infection by inducing the soybean plant roots to produce substances with nematode killing activity or inhibiting the nematode’s growth and development.” in L.53-56.

2. Response to comment: L. 59: not clear of how the bacteria inoculum were coated to the seeds.

Reply: We explain the methods how the bacteria inoculum were coated to the seeds in L.66-67 “Subsequently, the seeds were washed with 70% ethanol and 1% of the seeds were coated with Sneb545.”

3. Response to comment: L. 67: not clear of the seedling age or growth stage at the time of inoculation with SCN.

Reply: L.77-78 When two true leaves were grown from soybean, the root of soybean was inoculated with soybean cyst nematode.

4. Response to comment: L. 70: plants were assessed at 5 and 10 days after inoculation: not sure if such short period was enough for observing interactions between the bacteria and SCN

Reply: L79: After the previous observation on the development of nematode in soybean root under the microscope, we found that there were four larvae at 10 days after nematode infected soybean, so although the difference was only 5 days, the development state of nematode in these two time points was obviously different.

5. Response to comment: L. 81: samples of roots?

Reply: Yes. The sample is root tissue of the soybean.

6. Response to comment: L. 74: ‘Liaodou15’ here, but ‘Liao15’ in L. 58. Use one and be consistent throughout the manuscript

Reply: We have changed the L.84‘Liaodou15’ into‘Liao15’.

7. Response to comment: L. 77-78: last sentence on data analysis can be removed to the “Data analysis” section

Reply: We removed the L.77-78 last sentence on data analysis to the “Statistical analysis” section in L.120-121 “The nematode infection and mortality data were examined using Student’s t-test in SPSS 17.0 (IBM, USA) and a P-value of < 0.05 was considered statistically significant.”

8. Response to comment: L. 107-114: put this section last, i.e. before the ‘Results’ section to include all acquired data in this study in addition to the spectrometry data, such as ANOVA for nematode numbers etc.

Reply: We put this section ‘Statistical analysis’ before the‘Results’ section in L.120-128.

9. Response to comment: L. 129: significant level? The statistical analysis was not mentioned in the M&M section.

Reply: We added the statistical analysis in the ‘Statistical analysis’ section in L.121-123.

10. Response to comment: L. 130-131: delete this sentence and move it to Discussion section.

Reply: We delete this sentence and move it to Discussion section in L.162-163.

---

## [Decision Letter · Decision Letter 1]

30 Jun 2020

PONE-D-20-07320R1

Bacillus simplex treatment promotes soybean defence against soybean cyst nematodes: A metabolomics study using GC-MS

PLOS ONE

Dear Dr. Kang,

Thank you for submitting your manuscript to PLOS ONE. After careful consideration, we feel that it has merit but does not fully meet PLOS ONE’s publication criteria as it currently stands. Therefore, we invite you to submit a revised version of the manuscript that addresses the points raised during the review process.

We look forward to receiving your revised manuscript.

Kind regards,

Hon-Ming Lam, Ph.D.

Academic Editor

PLOS ONE

Additional Editor Comments (if provided):

This manuscript needs a thorough English polishing. The authors may consider to get professional English editing service.

Reviewers' comments:

Reviewer's Responses to Questions

**Comments to the Author**

1. If the authors have adequately addressed your comments raised in a previous round of review and you feel that this manuscript is now acceptable for publication, you may indicate that here to bypass the “Comments to the Author” section, enter your conflict of interest statement in the “Confidential to Editor” section, and submit your "Accept" recommendation.

Reviewer #2: (No Response)

2. Is the manuscript technically sound, and do the data support the conclusions?

Reviewer #2: Yes

3. Has the statistical analysis been performed appropriately and rigorously? 

Reviewer #2: Yes

4. Have the authors made all data underlying the findings in their manuscript fully available?

Reviewer #2: Yes

5. Is the manuscript presented in an intelligible fashion and written in standard English?

Reviewer #2: Yes

6. Review Comments to the Author

Reviewer #2: The authors have attended all comments and carefully revised the manuscirpt. Higher resolution / better quality of figures are required for the publications. The writing language has been largely imporved in the R1. However, there are still many language problems requiring further improvements. Let's take the first paragraph of the Introduction section as examples:

L. 33: change "Soybean has a high protein and oil content" to "Soybean seeds contain high protein and oil content".

L. 34 "which place high importance on its production yield" is not clear.

L. 36 "can hinder its growth and production": not clear of "its" - just use "soybean", and can be changed to "can hinder the growth and production of soybean".

L. 39-40: "Plant growth-promoting rhizobacteria (PGPR) colonises the rhizosphere of" should use "colonise".

L. 42-44: replace "Our past studies have discovered the Sneb545 strain of Bacillus simplex that can enhance soybean resistance to SCN" with "Our past studies have discovered that the Sneb545 strain of Bacillus simplex enhanced soybean resistance to SCN"

.....

7. PLOS authors have the option to publish the peer review history of their article (what does this mean?). If published, this will include your full peer review and any attached files.

Reviewer #2: **Yes: **Yinglong Chen

---

## [Author Response · Author response to Decision Letter 1]

9 Jul 2020

Response to Reviewers

Dear Editors and Reviewers:

Thank you for your letter and for the reviewers’s comments concerning our manuscript entitled “Bacillus simplex treatment promotes soybean defence against soybean cyst nematodes: A metabolomics study using GC-MS”(PONE-D-20-07320). Those comments are all valuable and very helpful for revising and improving our paper, as well as the important guiding significance to our researches. We have studied comments carefully and have made correction which we hope meet with approval. Revised portion are marked in red in the paper. The main corrections in the paper and the responds to the editors and reviewer’s comments are as flowing:

Responds to the reviewer’s comments:

Reviewer 2:

1. Response to comment: Higher resolution/better quality of figures are required for the publications.

Reply: We enhanced the resolution to 300*300 of the figures .

2. Response to comment: change "Soybean has a high protein and oil content" to "Soybean seeds contain high protein and oil content".

Reply: We have changed the sentence to “Soybean seeds contain high protein and oil content” in L.33

3. Response to comment: L. 34 "which place high importance on its production yield" is not clear.

Reply: We have changed the sentence to “This places a high importance on efficient production of soybean with a high yield” in L. 34.

4. Response to comment: L. 36 "can hinder its growth and production": not clear of "its" - just use "soybean", and can be changed to "can hinder the growth and production of soybean".

Reply: We have changed the sentence to “can hinder the growth and production of soybean” in L.37.

5. Response to comment: L. 39-40: "Plant growth-promoting rhizobacteria (PGPR) colonises the rhizosphere of" should use "colonise".

Reply: We have changed the word to “colonise” in L.41.

6. Response to comment: L. 42-44: replace "Our past studies have discovered the Sneb545 strain of Bacillus simplex that can enhance soybean resistance to SCN" with "Our past studies have discovered that the Sneb545 strain of Bacillus simplex enhanced soybean resistance to SCN"

Reply: We have changed the sentence to "Our past studies have discovered that the Sneb545 strain of Bacillus simplex enhanced soybean resistance to SCN" in L.43-45.

7. We also modified the other language problems in the ‘Revised Manuscript with Track Changges’.

---

## [Editor Report · Decision Letter 2]

22 Jul 2020

Bacillus simplex treatment promotes soybean defence against soybean cyst nematodes: A metabolomics study using GC-MS

PONE-D-20-07320R2

Dear Dr. Kang,

We’re pleased to inform you that your manuscript has been judged scientifically suitable for publication and will be formally accepted for publication once it meets all outstanding technical requirements.

Kind regards,

Hon-Ming Lam, Ph.D.

Academic Editor

PLOS ONE
---

## [Editor Report · Acceptance letter]

28 Jul 2020

PONE-D-20-07320R2 

Bacillus simplex treatment promotes soybean defence against soybean cyst nematodes: A metabolomics study using GC-MS 

Dear Dr. Kang:

I'm pleased to inform you that your manuscript has been deemed suitable for publication in PLOS ONE. Congratulations! Your manuscript is now with our production department. 

Kind regards, 

on behalf of

Dr. Hon-Ming Lam 

Academic Editor

PLOS ONE